# Chitosan, Gelatin, and Collagen Hydrogels for Bone Regeneration

**DOI:** 10.3390/polym15132762

**Published:** 2023-06-21

**Authors:** Karen Guillén-Carvajal, Benjamín Valdez-Salas, Ernesto Beltrán-Partida, Jorge Salomón-Carlos, Nelson Cheng

**Affiliations:** 1Departamento de Corrosión y Materiales, Instituto de Ingeniería, Universidad Autónoma de Baja California, Blvd. Benito Juárez and Normal s/n, Mexicali 21280, Baja California, Mexico; kguillen@uabc.edu.mx (K.G.-C.); benval@uabc.edu.mx (B.V.-S.); jsalvador@uabc.edu.mx (J.S.-C.); 2Laboratorio de Biología Molecular y Cáncer, Instituto de Ingeniería, Universidad Autónoma de Baja California, Blvd. Benito Juárez y Calle Normal s/n, Mexicali 21280, Baja California, Mexico; 3Magna International Pte Ltd., 10 H Enterprise Road, Singapore 629834, Singapore; nelsoncheng@magnachem.com.sg

**Keywords:** bone regeneration, osteoblast–hydrogel interaction, hydrogel cross-linking, drug delivery, hydrogel biomaterials

## Abstract

Hydrogels are versatile biomaterials characterized by three-dimensional, cross-linked, highly hydrated polymeric networks. These polymers exhibit a great variety of biochemical and biophysical properties, which allow for the diffusion of diverse molecules, such as drugs, active ingredients, growth factors, and nanoparticles. Meanwhile, these polymers can control chemical and molecular interactions at the cellular level. The polymeric network can be molded into different structures, imitating the structural characteristics of surrounding tissues and bone defects. Interestingly, the application of hydrogels in bone tissue engineering (BTE) has been gathering significant attention due to the beneficial bone improvement results that have been achieved. Moreover, essential clinical and osteoblastic fate-controlling advances have been achieved with the use of synthetic polymers in the production of hydrogels. However, current trends look towards fabricating hydrogels from biological precursors, such as biopolymers, due to the high biocompatibility, degradability, and mechanical control that can be regulated. Therefore, this review analyzes the concept of hydrogels and the characteristics of chitosan, collagen, and gelatin as excellent candidates for fabricating BTE scaffolds. The changes and opportunities brought on by these biopolymers in bone regeneration are discussed, considering the integration, synergy, and biocompatibility features.

## 1. Introduction

Bones consist of highly vascularized tissue, capable of auto-regenerating as part of a reparation process in response to injury, as well as during skeletal development and continuous remodeling throughout adulthood [1,2]. Contrasting with other tissues, most bone injuries (such as fractures) heal without forming scar tissue and are indistinguishable from the adjacent non-injured bones [2,3]. Currently, millions of patients suffer from bone defects due to trauma, bone disease, congenital malformations, and cancer [4]. The repair of significant bone defects is a great orthopedic challenge worldwide due to the difficulty of conducting and restoring new bone. [5]. Moreover, medical factors such as age, gender, lifestyle, and preexisting conditions influence the risk of fracture and complications arising during the recuperation process [6,7,8]. A recent study on the global burden of morbidity suggested that approximately 178 million people (53% male and 47% female) worldwide suffered from bone fractures in 2019, exemplifying an increase of roughly 34% since 1990 [9]. In fact, approximately 2.2 million bone grafting surgeries have been performed globally at the expense of about USD 2.5 billion per year, and that number is gradually increasing due to the aging population [10,11].

Despite the advanced methods available to treat bone damage and fractures, autografts are considered the “gold standard” since they provide optimal tissue acceptance and controlled osteogenesis. Nevertheless, autografts have significant downfalls; they can result in postoperative complications, such as hernias; blood loss; nerve damage; necrosis; and, more critically, systemic infections [3,12,13,14,15,16]. Surgeons have the alternative option of using allografts, although there is a risk of immunogenic reactions and viral transmissions [13,17,18,19]. Nonetheless, several options have been tested: for instance, the generation of synthetic prostheses capable of offering the same mechanical properties as bone. However, in the long term, prostheses can present the same complications as autologous and allogenic implants [20]. On the other hand, the application of natural hydrogels in BTE has been gathering more attention due to the advantages of designing matrix polymeric biomaterials loaded with osteogenic-inducing molecules. By controlling hydrogel synthesis and matrix properties, we can regulate the release profile and the mechanical parameters, making them ideal for BTE scaffold design [21]. Many significant advances have been achieved using natural polymers to construct hydrogels due to the precise control over chemical structures, low batch variability, and facile sourcing [22]. Recent trends have included the fabrication of hydrogels from biological macromolecules to introduce a specific biofunctionality, and this has promoted cell–material interactions inherent to the given hydrogels [23].

We consider the current demand for the development of novel technologies and highly engineered procedures for fabricating functionalized BTE scaffolds (e.g., 3D ex vivo and in vivo bioprinting, electrospun fibers, injection molding, injection of implants, and microfluidics), as well as the role of physical and chemical cross-linking, among other strategies. The natural hydrogel materials discussed in this paper consist of: chitosan, collagen, and gelatin, paying particular attention to the drug delivery properties, osteoblasts’ growing functionality, and, more importantly, the new advances for BTE applications. The design, gelling control, cross-linking chemistry, controlled release, and osteoblast growth are analyzed based on hydrogel’s structure properties. It is important to highlight that chitosan, collagen, and gelatin hydrogels are attractive options for developing contemporary biomaterials for BTE.

## 2. Hydrogels

Hydrogels are 3D cross-linked polymeric networks capable of imbibing large amounts of water (Figure 1). Moreover, the hydrogel polymeric matrix structure allows for the diffusion of diverse molecules, such as drugs, active molecules, growth factors, nanoparticles, and more. On the other hand, we can control the chemical and molecular interactions of the polymeric chains guiding the biological behavior from the cellular level. Furthermore, the polymeric network can be molded into different arrangements and sizes, following the structural characteristics of repairing tissue defects. Therefore, they can provide constructive microenvironments suitable for controlled cell growth [20,21,24,25,26]. It is interesting to emphasize that hydrogels can show versatile control of physical properties according to the exposed environmental conditions. For instance, hydrogels work as soft materials that can form solid structures (after dryness) that diverge in terms of mechanical properties, providing the capacity to generate solid scaffolds. In contrast, hydrogels can absorb significant amounts of water and preserve humid environments without necessarily decomposing or degrading their structural architecture [27]. Their high water content makes hydrogel materials highly permeable and porous, allowing oxygen and nutrients to diffuse quickly and resulting in a balanced interconnected microenvironment that can stimulate a guided cellular fate [28].

## 3. Hydrogel Classification

The classification of hydrogels (Figure 2) follows physicochemical properties, such as biochemical and biophysical response, synthetic methods, precursors, ionic charge, degradability, and cross-linking degree [29,30]. Inherently, the physically synthesized hydrogels show reversible cross-linking matrixes mainly characterized by coordinated electrostatic interaction, which forms Van der Waals forces and hydrogen bonds. Meanwhile, the chemically developed hydrogels are characterized by permanent and irreversible cross-linking bonds that require high energy to alter the matrix configuration [29]. On the other hand, the type of precursor material can also classify hydrogels as either natural or synthetic. It is generally considered that natural hydrogels are more biocompatible and bioactive than their synthetic counterparts. However, synthetic hydrogels promote controllable, mechanical, and degradable properties over naturally sourced polymers [21].

Natural hydrogels are composed of biopolymeric sources derived from animals and plants, which can be classified into two categories: polysaccharides and polypeptides. In the polysaccharides group, the most commonly used polymers are chitosan and alginate-based. On the other hand, polypeptides such as collagen and gelatin are mainly applied as supportive and guiding scaffolds for BTE. Therefore, biopolymers can incorporate attractive characteristics mandatory for functional biomaterials, including their chemical composition for cellular interaction and controlled degradation. Interestingly, the cross-linking degree of biopolymeric hydrogels plays an essential role in assembling (e.g., charge) due to the specific functional groups of certain biopolymers, or in interpolymer cross-linking due to physical and chemical modifications [31].

Biopolymers share similar components to the extracellular matrix (ECM), showing good biocompatibility, low immune response, and nearly null cytotoxicity compared to different synthetic polymers. Additionally, biopolymers can promote cellular adhesion, proliferation, and regeneration of bone-forming cells (osteoblasts) [32,33,34,35]. Therefore, from a biophysical point of view, hydrogels resemble many virtual properties of natural tissues. The morphological characteristics of the polymers allow for the exchange of substances, conducting cell adhesion in the initial stage and bone growth in the follow-up stage. It has been substantially demonstrated that cells are readily suspended in hydrogels and that the viability of encapsulated cells in the biopolymeric matrix can be largely preserved [36,37]. Thus, natural hydrogels are also biodegradable, providing initial support for the promotion of cellular adhesion. They degrade as cell populations grow and mature, changing the microenvironment and substituting with newly regenerated tissue [25].

## 4. Hydrogels for Bone Regeneration

The application of hydrogels in tissue engineering and regenerative medicine, particularly BTE, has attracted increasing attention due to the osteogenic drug delivery benefits that involve the polymeric matrix [21]. Important advances have been achieved using hydrogels based on synthetic polymers due to the inherent biocompatible properties of their natural source [22]. Recent trends have included the fabrication of hydrogels from biomacromolecules to introduce specific and inherent biofunctionality to hydrogels [23]. Furthermore, these hydrogels have demonstrated excellent integration with the surrounding tissues, avoiding the complex process of surgical removal due to failing response and reducing the possibility of inflammatory side effects [38]. These polymers can be tailored to obtain the desired geometry for implantation or injection. Moreover, we can easily control the degradation, porosity, and release profile by altering the method and degree of cross-linking [32]. Considering these parameters, hydrogels successfully provide structural support by simulating the natural tissue environment while offering a conductive regenerating scaffold for defective or imperfect sites. Thus far, allowing the bone to carry out its healing mechanism is imperative, as osteoblasts can adhere both on the surface of the hydrogel and within the hydrogel’s pores, ultimately leading to differentiation and maturing of the proliferating cells (Figure 3) [39].

Considering the above-stated information, we can postulate that hydrogel formulations, when used for bone regeneration, must meet specific standards when used as an implanted scaffold, or, in some circumstances, as an injectable system [40]:No cytotoxic and no immunogenic response, in order to avoid a chronic and non-regulable inflammatory reaction;Osteoinductive, osteoconductive, osteogenic, and osteocompatible qualities for better bone anchorage and regeneration;Mimicking the natural ECM at the implant site;Degradable by different enzymes or environmental molecules, leaving sufficient space for new bone formation;Resistant and stable during sterilization;Controlling the size and interconnection of the pores to optimize the characteristics of drug release, cell growth, oxygen diffusion, and nutrient exchange;Patient-friendly injectable form to reduce pain and simplify the administration process.

The structures of natural polymers should be similar to the ECM, providing comparable mechanical stability and bone integrity to prevent chronic inflammatory or immune responses. Additionally, the physical and chemical characteristics can increase the materials’ bioactivity, strength, and toughness in hydrogel applications [41]. However, the challenges related to controlled release or drug encapsulation still require further investigation [32]. Expanding the application of hydrogels in bone regeneration demands continuous formulation and improvement of the methods of preparation, as well as the development of in vitro and in vivo tests to enhance biocompatibility and osteoconductive capabilities.

## 5. Chitosan as a Carbohydrate Material for Hydrogels

Chitosan (CS) is produced by chemical or enzymatic deacetylation of chitin isolated from crustaceans, insects, and some microorganisms [20,42,43]. CS consists of glucosamine and N-acetyl-D-glucosamine repeated units linked by covalent β-1-4-glucosidic bonds that are disrupted, especially by enzymatic reactions under biodegradability conditions. The degradation rate and hydrophilicity of CS are influenced by the degree of deacetylation, which can vary from 30% to 95% [44]. Moreover, CS has been reported to be secure, biocompatible, osteoconductive, antibacterial, and immune-modulatory, and promotes bone formation in vivo [45,46,47,48]. In addition, the cationic nature of CS imparts hemostasis and mucosal adherence, and is a great drug manager in hydrogels [49,50]. It is important to highlight that the physicochemical properties of CS are mainly modulated by the degree of deacetylation and the molar mass weight [51]. Therefore, the degree of deacetylation is proportional to the solubility, viscosity, biocompatibility, mucosal adherence, and antibacterial and hemostatic activity. Similarly, CS’s crystallinity and biodegradability decrease when the degree of deacetylation is reduced. Meanwhile, the biodegradability and antioxidant activity are in line with the molar mass and tridimensional configuration [52].

Interestingly, the unique properties of CS arise mainly from the presence of amino functional groups. CS cannot be dissolved in organic solvents, but only in acid media or dilute acidic solutions. Thus, the acidic solutions conduct the protonation of the amino groups, converting CS ions into cationic polyelectrolytes. According to Rinaudo et al. and Wang et al. [53,54], the CS solubility transition occurs between pH values of 6.0 and 6.5, directly impacting the control of the “charged state” (positively charged and uncharged) in the processes of osteoblast adhesion, bone ingrowth, and subsequent mineralization [52,55]. The presence of amino groups also enables the formation of ionic complexes, for example, with metal ions [47,56]. Similarly, once the polycation is formed, CS can form heteropolymers with negatively charged organic compounds, such as lipids, proteins, DNA, or poly(acrylic acid) [47]. Nonetheless, the amino groups can undergo reductive amination to obtain aldehyde groups [47,57]. Additionally, CS chains contain hydroxyl groups that can form covalent bonds and allow for the integration of various electrostatic internal interactions.

From a biomaterial perspective, CS is a fascinating natural polymer that exhibits a variety of valuable applications. CS can be applied for coating inorganic nanoparticles and can be implemented in film or membrane fabrication, in the configuration and controlled release of drugs, in antimicrobial properties, or combined with different regenerative materials [58,59]. Likewise, it can be used as a coating composite for magnetic materials and for synthesizing ecological nanoscavengers to improve the quality of water bodies [60]. Interestingly, cross-linked CS polymers can be classified as both physical and chemical hydrogels. [61]. The physical hydrogels present a three-dimensional network formed by unstable electrostatic bonds interacting with anionic and cationic molecules to stabilize the polymeric network [62]. Moreover, the physical CS gels have lower mechanical properties and reduced cytotoxicity, allowing for the possibility of regulating both the extension and the swelling rate more significantly than chemical gels. Thus, controlling the sol–gel transition by modifying physicochemical parameters, such as temperature, pH, and ionic strength, or by adding suitable counter ions, are key conditions for directing gel formation [63,64]. Furthermore, electrostatic and hydrophobic interactions with hydrogen bonds cooperate to form interlocked stable networks in the as-formed CS hydrogel. Similarly, the kinetics of self-assembly generally occur very quickly, although the process can be easily regulated by increasing the ionic strength or the cross-linker type and concentration [64].

## 6. Collagen as a Protein Structural Material for Hydrogels

Collagen is the principal ECM component of vertebrates, and is the most abundant protein in the animal kingdom. In the human body, more than 90% of collagen is type I, II, or III [65]. The collagen family has at least 28 members, and depending on their structure and organization, collagen can be grouped into [66]:Fibril-forming;Fibril-associated collagens;Network-forming collagens;Anchoring fibrils;Transmembrane collagens;Basement membrane collagen;Others with unique functions.

Among all types of collagen proteins, type I collagen (Figure 4) is the most frequent in the ECM, principally in tissues such as tendons and bones [17,34,67]. Interestingly, type I collagen improves mechanical strength and new bone remodeling through mineralization in mature bones [68]. However, according to Hermann Ehrlich, collagen types II and III are found significantly in other body parts, such as cartilage and soft tissues, respectively [69]. For instance, collagen type II accounts for 80% of the total collagen content in cartilage, and the amount of proteoglycan present depends on the cartilage type. It is present in a higher proportion in tendons, and can also form a homotrimeric molecule similar in size and biomechanical properties to type I collagen [66,67]. Type III is a homotrimer widely distributed in collagen I which contains tissues except for bone, tendon, and cartilage [67]. Nonetheless, collagen type I is the most important ECM protein, modulating maturation, deposition of hydroxyapatite, and biochemical functions related to bone-forming cell behavior.

Tropocollagen is the basic structural unit of collagen proteins, and consists of three intertwined peptide chains of approximately 1000 amino acids. Tropocollagen forms solid aggregating staggered sets of tropocollagen molecules. Each triple helix has a length of 300 nm, which is, interestingly, the same length as that reported for the structural unit of chitin [70]. Furthermore, the individual collagen chain is composed of a repetitive sequence, Glycine-X-Y [70,71], where X and Y are usually proline and hydroxyproline, respectively [72]. It is important to highlight that the tight wrapping of the triple helix chains provides collagen with a higher tensile strength than that of steel wire of equal cross-sections. Collagen provides strength and confers a shape while allowing for flexibility and movement. Meanwhile, it responds to tension in soft tissues and provides a platform for mineralization in hard tissues subject to compression [69,71]. Likewise, collagen’s ubiquity also improves efficiency and economic isolation from tissues such as the skin, tendons, and pericardium. Collagen is usually extracted from bovine, porcine, or rat specimens, but also from marine species and recombinant sources [72]. Similarly, it is widely used as a biomaterial for tissue regeneration, such as with hydrogels.

Interestingly, collagen is a highly biocompatible material, providing the ideal environment for cell adhesion and proliferation, principally for osteoblasts [73]. This makes it a great candidate for tissue regeneration [70]. Recent studies have been conducted on collagen’s role in biomineralization, both in vivo and in vitro. Biocompatibility and safety are achieved due to biological characteristics such as biodegradability and weak antigenicity, highlighting collagen as one of the leading resources in biomedical applications [69]. Furthermore, collagen’s ability to form intrafibrillar and interfibrillar cross-links can be exploited to create hydrogels with a range of controllable mechanical properties closely related to those of different types of bones [70]. Similarly, collagen type I is the most frequently used for biomedical applications. This emphasizes the importance of studying and developing hydrogel materials using this resource [69,70,73,74,75].

## 7. Gelatin Material for Hydrogels Development

Gelatin is a hydrolyzed and denatured form of collagen extracted from the skin and bones of animals (often from porcine skin) by acid or alkaline treatment, followed by a thermal protein separation process [76,77]. The central part of the triple helix structure of native collagen is denatured during the production of gelatin; however, the chemical structure of gelatin remains similar to that of collagen [77]. Gelatin conserves the Glycine-X-Y amino acid repeat sequences of the collagen base’s primary structure [78]. Similarly, gelatin also includes the RGD (Arg-Gly-Asp) sequence, a cell adhesion site that binds integrins in the gelatin web matrix. Incorporating gelatin (and, thus, RGD) into biomaterials has improved cell integration and tissue repair in different applications [79]. Gelatin can also be degraded by proteases, such as collagenase and metalloproteases, thus suggesting a close biochemical relationship between gelatin and collagen [80].

Gelatin has been widely used in the food and pharmaceutical industries as a stabilizer, thickener, texturizer, and emulsifier. In clinical applications, gelatin has been mainly explored as an ingredient in capsules, tablets, and sponges [77,81]. Gelatin particles have been widely applied as hemostatic agents to fix cartilage and bone defects [82], also highlighting the advantages of gelatin’s biological functionality (RGD sequence) for tissue engineering [83]. Therefore, it is important to describe the benefits of gelatin and methods of conjugating it with different functional groups to form copolymeric hydrogels and, thus, achieve more biological and mechanical properties [84,85]. Gelatin gels are typical thermoreversible gels obtained above 40 °C and gelled when the temperature drops below the gelling temperature (30 °C). Gelation is caused by partially restoring the original interconnected conformation of the triple helices present in collagen to form three-dimensional networks [86,87]. The gel formation is mainly due to hydrogen bonds, and is called “physical gel” [87,88,89].

## 8. Application of Chitosan, Gelatin, and Collagen Hydrogels in BTE

In a previous study, Savaranan et al. [90] applied an injectable thermosensitive hydrogel based on CS and glycerophosphate (GP) to heal and regenerate bone defects. The authors showed that adding graphene oxide (GO) improved the mechanical properties of the hydrogel. Furthermore, the work illustrated that both synthesized hydrogels exhibited optimally sized interconnective pores, and the presence of GO did not significantly alter the pores’ geometry (Figure 5A–C). The results suggested that the characteristic pores permitted cell infiltration, new tissue ingrowth, nutrient transport, and active ingredient diffusion, improving the scaffold functionality [91]. On the other hand, the results suggested that adding GO improved the physicochemical properties (protein adsorption, swelling capacity, and control of the degradable behavior of the hydrogel) while maintaining the thermosensitive property. Likewise, the CS/GP/GO hydrogel was biocompatible with mesenchymal stem cells, resulting in continuous adhesion and proliferation of the cells among the hydrogels (Figure 5D).

During the biomineralization process, a hydroxyapatite layer is formed on a biomaterial-based construct, following a sequential deposition mechanism of calcium and phosphate ions from the surrounding body fluids to aid in osteointegration. This interesting process of interaction is conducted to deposit calcium phosphate, which increases with the addition of GO. Similarly, cellular differentiation was measured, which indicated that the GO improved calcium deposition and osteogenesis. It is important to highlight the doping of hydrogels to obtain composite scaffolds, such as those of GO, and, therefore, to promote the material’s bone-forming functionality.

Recently, a CS-gelatin-nanohydroxyapatite (CSG/nHaP) composite scaffold with a pore size of 100–180 µm showed improved adhesion and growth of mouse calvaria pre-osteoblast cells (MC3T3-E1) compared to nHAp alone [92]. Similarly, a recent work described its osteogenic action when lyophilized and cross-linked with glutaraldehyde or genipin CS/gelatin scaffolds. The research demonstrated that bone regeneration was tailored by introducing ECM into a mice model, resulting in a minimal inflammatory reaction. The immunoassay for the expression of cell surface markers showed a homogeneous cell population of positive CD90, endoglin, and Ecto-5′-nucleotidase, with negative results for CD45 and mucosialin. Thus, the scaffold permitted osteogenic cellular differentiations to occur [93]. In another research paper [5], the researchers designed a scaffold using chemically sulfonated graphene oxide. This scaffold showed well-controlled drug delivery and bone regeneration due to the insertion of porous network structures, which resulted in weak Van der Waals interaction and increasing the interlayer spacing by the sulfonated groups in GO. In the same way, the sulfonated groups interacted with the drug, allowing its release to occur more slowly. According to the cell viability results, superior biocompatibility was obtained using GO sulfonated materials compared to their non-modified counterparts. Similarly, in vivo studies demonstrated that these scaffolds had the potential to regenerate bone tissue more quickly, and without side effects, compared to pure CS scaffolds.

Interestingly, Nguyen et al. [94] demonstrated that adding TEMPO (2,2,6,6-tetramethylpiperidine-1-oxyl radical)-oxidized cellulose nanofibers (TOCNF) to CS increased the gelation and porosity, and improved the biocompatibility, both in vitro and in vivo. The authors suggested that TOCNF resulted in a thermosensitive injectable hydrogel. Moreover, the composite hydrogel underwent a sol–gel transition at body temperature, indicating that it was a transparent liquid solution (Figure 6). Moreover, the experimental results illustrated that CS/TOCNF improved the biocompatibility of MC3T3-E1 preosteoblast cells and L929 fibroblasts compared to the control hydrogel. The implanted hydrogel was also subjected to immunofluorescence staining to analyze the receptors associated with the activated macrophages surrounding the implant. The results showed an initial inflammatory response with a presence of activated macrophages after 2 weeks, which was indicative of the healing process.

Rami et al. [95] reported that the biological responses were significantly different due to properties in the hydrogel derived from the primary polymer. For example, the polymer concentration, the gelation process, and the degree of acetylation (DA) in CS homopolymers are essential parameters that demand special attention (Figure 7). It is important to highlight that the CS-derived hydrogels with low DAs were more elastic, while, on the contrary, those with high DAs were their softer counterparts. Furthermore, these mechanical properties of low-DA groups promote the adherence and spreading of human bone marrow mesenchymal stem cells (hBMSCs) and human progenitor-derived endothelial cells (hPDECs) on the CS hydrogel surface. Meanwhile, CS polymers with high DAs are unsuitable for human cell culture application. The current evidence offers a promising and innovative way for the design process of hydrogel materials with tunable properties for BTE and regenerative medicine.

It has been reported that inorganic content in polymeric composite materials can modulate the differentiation process of mesenchymal stem cells, while showing a positive effect on the expression of anti-inflammatory cytokines [96]. Recent work has suggested that a sol/gel, followed by lyophilization of a CS hydroxyapatite scaffold, can improve osteoblast maturation in a size-dependent manner. Interestingly, Soriente et al. showed that small-sized hydroxyapatite particles promoted faster osteoblast calcification than their larger counterparts. Moreover, Guo et al. [97] fabricated collagen/CS electrospun nanofiber membranes. The results showed higher tensile strength, a more stable degradation rate, and better in vivo results in the repair of cranial bone defects than electrospun membranes based on collagen alone. In this work, the authors proposed that the nanofiber configuration modulates the mechanical properties adequate for adjusting to the required scaffolding applications.

Previously, Babaei et al. [98] synthesized a new type of bone replacement hydrogel using the in situ precipitation method. The authors mixed a gelatin solution with a CS suspension containing Ca^2+^ and PO_4_^3−^ ions. The CS and gelatin were cross-linked by using glutaraldehyde under alkaline conditions, and calcium phosphate (CaP) crystals were precipitated within the CS-gelatin matrix, resulting in a CS-Gel/CaP composite. According to the authors’ interpretation, the formation mechanism of the CaP nanoparticles in the polymeric matrix consisted of 3 steps (Figure 8, top). A layer of Ca^2+^ ions was formed on the surface of CS to allow for the interaction of amino and carbonyl groups, then another layer of PO_4_^3−^ ions was attracted to the Ca^2+^ layer through electrostatic interaction. By adding gelatin, the free Ca^2+^ ions interacted with functional groups of gelatins and permitted other layers to form, as the PO_4_^3−^ ions joined the calcium ions. Finally, the amino groups in CS and the carbonyl groups in the gelatin interacted through glutaraldehyde, forming the final matrix. Far more interesting were the SEM micrographs (Figure 8, down) of the hydrogel, which revealed the formation of calcium phosphate with high dispersion in the polymer matrix due to the polymer network’s density. Moreover, the SEM suggested that the polymer network’s density controlled the formation of calcium phosphate in the polymeric matrix. An important point to highlight is that, according to their studies, when a higher-density polymeric matrix was used, the crystals of the obtained inorganic phase were smaller.

Previously, Guo et al. [99] fabricated a gelatin/CS composite hydrogel containing titanium oxide (TiO_2_) nanoparticles to control the acceleration of bone fracture healing. Interestingly, the hydrogel was connected in a tubular form by adding TiO_2_. The gelatin acted as a stabilizing agent; meanwhile, the TiO_2_ nanoparticles spread in the matrix, improving its heating resistance and providing support and several active sites for the nucleation of polymer units. Moreover, the CS/gelatin/TiO_2_ hydrogel showed a higher expression of osteocalcin and F-actin markers than CS/gelatin hydrogel, indicating the formation of mature osteoblasts.

Peter et al. [100] synthesized CS–gelatin composite scaffolds loaded with bioactive glass–ceramic nanoparticles (nBGC), using the sol-gel process as well as freezing and freeze-drying. The resulting bioactive glasses were osteoconductive and biodegradable, with potential applications for bone repair. Similarly, the addition of nanoparticles regulated the scaffolds’ density and degradation. From PART 1, Figure 9a,b, we can analyze the SEM micrographs of an apatite-like layer on the surface of CG/nBGC after 7 days and 14 days of biomineralization (c, d). From the chemical point of view, the EDS spectra of mineralized zones demonstrated the bioactive nature of the nanocomposite scaffolds due to the Ca/P ratio of 1.64 (e). In PART 2, Figure 9a,b, the MTT assay demonstrated that the scaffolds provided a healthy environment for cell proliferation, as sufficient microporosity was obtained for cell infiltration. These important responses suggest that the experimental hydrogels have potential applications in alveolar bone regeneration (PART 2, Figure 9a) [101].

In a recent study, Liu et al. [102] designed catechol–chitosan (CA-CS) hydrogels functionalized with zeolitic imidazolate frameworks and 8 and ZIF-8 (ZIF-8 NP) (CA-CS/Z) nanoparticles. The aim was to ensure an adequate blood supply, maintain stabilization of the bone transplant environment, enhance osteogenesis, and promote the bone regeneration conducted by the CA-CS/Z. Moreover, the hydrogel demonstrated satisfactory adhesion and antimicrobial activities. On the other hand, the ZIF-8 discharged from the hydrogels was also able to enhance the release and formation of osteocalcin, collagen I (COL1), and alkaline phosphatase (ALP) markers, improving the osteogenic differentiation of rat bone marrow mesenchymal stem cells (rBMSCs). It is important to highlight that this type of hydrogel could be an effective bone adhesive with multiple biofunctions, such as promoting vascularized osteogenesis. These remarkable properties are provided by the catechol groups that form strong covalent and noncovalent bonds, thus improving the adhesion of hydrogels, while the nanoscale ZIF-8 releases zinc ions, which plays an active role in osteogenesis, angiogenesis, and antibacterial processes. Moreover, combining these active components improves the mechanical properties of the chitosan hydrogel, making it a great candidate for BTE.

Recently, a novel product focused on preventing and eliminating periodontal and peri-implant disease of the gum (periodontitis) was developed using CS, and is marketed under the name of Periosan^®^. Interestingly, the contact between the dental implant and the alveolar bone (peri-implantitis) triggers the massive destruction of the gum and the alveolar bone following the total loss of the dental implant. This new technology was fabricated in the format of gels and membranes, using triclosan microcapsules as active ingredients, which were able to completely reduce *S. mutants*—bacteria responsible for the initial mechanism of peri-implantitis. Moreover, Periosan^®^ was shown to be capable of reducing the microorganisms obtained inside the implant of a diabetic patient, which led to a significant decrease in the growth of microorganisms inside the implant [59].

On the other hand, gelatin-based hydrogels incorporating stromal cell-derived factor-1 (SDF-1) and bone morphogenetic protein-2 (BMP-2) showed that the growth factor’s combined release induced bone regeneration more significantly than a single release. This work provides new bases for understanding and studying controlled release systems and the order of release kinetics for the purpose of delivering combined growth factors [103]. In a previous study, Mir Hamed Nabavi et al. [34] promoted bone regeneration using type I collagen hydrogels containing tacrolimus (Tac). Interestingly, Tac is a macrolide-type antibiotic that can act as an immunosuppressant [104]. Furthermore, recent studies have claimed that Tac may enhance osteogenic differentiation by activating BMP receptors, although its mechanism remains unclear [105]. Therefore, the in vitro and in vivo test results indicated that the collagen hydrogel containing 1000 μg/mL Tac was adequate for cell proliferation and bone healing. Similarly, Han et al. [106] synthesized a gelatin hydrogel with silver nanoparticles (AgNPs) to improve bone regeneration and fracture treatment. The results indicated that the AgNPs/Gel hydrogels which were composited were not harmful to osteoblasts and exhibited higher cell viability, combining the antibacterial benefits provided by the AgNPs.

Likewise, Arakawa et al. [107] created methacrylate glycol chitosan (MeGC) and semi-interpenetrating collagen (Col) hydrogels with photoinitiator capacity under visible light. The study showed that including Col improved the physical properties due to the formation of a tight network structure between the cross-linked CS and Col. Also, Col enhanced the cell propagation, proliferation, and osteogenic differentiation of BMSCs cultured both on the surface and within the hydrogels by day 1. Therefore, the authors highlighted the high potential of this hydrogel to act as a scaffold in BTE. Moreover, Kaur et al. [108] developed injectable chitosan–collagen hydrogels (CS/Col) using carboxylic acid-functionalized single-walled carbon nanotubes (COOH-SWCNTs) as integrators. In addition, they used sodium β-glycerophosphate (β-GP) salt as a cross-linker and thermoresponsive initiator of the sol-gel transition at a physiological temperature of 37 °C. The combination of these components allowed for optimal thermoresponsive and injection properties to be achieved, significantly improving the mechanical properties. In addition, the degradation and swelling were dependent on the composition. The dual charge nature of the hydrogel components allowed for the formation of HAp on the surface of the hydrogel after only 1 day of incubation. Far more importantly, the hydrogels were non-toxic, and they facilitated cell proliferation and osteogenic differentiation compared to pure CS/Col.

Finally, Gharati et al. [109] investigated the potential capacity of collagen hydrogel nanocomposites in combination with 2% strontium (Co/BGSr2%). The hydrogels were evaluated using both in vitro and in vivo studies for 56 days. In all the studies (radiographical and histopathological scoring at different times), better results were highlighted in the Co/BGSr2% + MSC (mesenchymal stem cells) group. The highest expression level of osteocalcin was detected in Co/BGSr2% + MSCs, especially by the fourth week post-transplantation.

New techniques have been under investigation for the purpose of developing novel technologies for BTE scaffolding. Keriquel et al. [110] used laser-assisted bioprinting (LAB) to reconstruct MSCs/Col/nHA bone substituents in situ. The authors obtained promising results, highlighting the impact of promoting MSC arrangement and subsequent differentiation on guiding bone regeneration. Similarly, Demirtas et al. [111] showed that chitosan-nHA-based bio-ink allowed for efficient MC3T3-E1 viability, proliferation, mineralization, and cell morphology in the hydrogels produced by bioprinting compared to those obtained with alginate/nHA. Thus, a new synthetic strategy was discovered for the treatment of bone defects.

In another work, Mohandesnezhad et al. [112] synthesized a chitosan/alginate/hardystonite (CS/Alg/HD)-based hydrogel scaffold by employing an extrusion-based 3D printing technique. The authors suggested that adding HD would improve the mechanical properties and increase the cell viability by promoting in vitro cellular attachment. Nonetheless, HD incorporation decreased the degree of swelling, thus increasing the degradability.

Considering the extended ability of Col, CS, and gelatin to act as base materials for hydrogel configurations, as well as their desired mechanical parameters and excellent ability to form composite nanomaterials, they were ideal materials for conducting the bone regeneration process applied in BTE. The application of chitosan, collagen, and gelatin biomaterials for BTE is summarized in Table 1.

## 9. Prospects and Limitations

The study, development, and implementation of biomaterials for BTE should consider integrating, in a synergic manner, the widely-demanded properties needed to achieve a functional bone scaffolding material. Interestingly, when designing novel scaffolds of naturally occurring sources, we should consider the physicochemical and cellular interactions that the hydrogel will carry out. For example, a BTE scaffold should direct cell adhesion and proliferation, always paying particular attention to the capability to integrate and form new bone by avoiding toxic events. Moreover, it is important that the matrix shape can be remodeled and structured following the undergoing bone defect. This physical advantage promotes the application of hydrogels, which can be modulated by selecting the type of co-polymer and the cross-linking behavior. On the other hand, hydrogels are versatile vehicles for the in situ administration of drugs, as well as organic, inorganic, and nanostructured molecules, with inherent pharmacological and bone-active properties, as discussed in this review. Regarding drug delivery applications, the synthesis of hydrogels to achieve controlled, guided, and stimulus-responsive biodegradability is orchestrated by the parameters mentioned earlier and the porosity level of the polymeric matrix.

Natural hydrogel-derived biopolymers face important challenges from a scaling point of view. The interaction of their components can result in different polymeric responses, which points out that some hydrogels are easily obtained in a laboratory environment (setting). However, in industrial scaling (setting), they may need a more controlled and elaborate procedure that can make the product more expensive in the best-case scenario. In the worst-case scenario, their development will not be possible. Moreover, the use of materials of natural origin can be variable due to their sources, such as animals, which could be more sustainable and hinder the extraction of Col and gelatin. On the other hand, we should consider the mechanical and cohesion properties within the hydrogel using cross-linking agents, which can be expensive. For instance, the cross-linker genipin, which is used in gelatin and collagen-based hydrogels, can overprice the process of industrial scale-up production. Another aspect to consider is the polymeric precursor’s solubility and the medium in which the biopolymers can be solubilized. Furthermore, developing hydrogels with drug-release properties can achieve different release profiles according to the load concentration and the degradation capacity of the matrix, which is limited in biopolymeric materials. Finally, special attention should be paid to the low bioadhesive capacity in humid environments such as the human body, a strict parameter that is required during bone integration.

Future studies should focus on obtaining high-quality natural biomaterials that can reduce or eradicate the previously described difficulties without endangering the properties or capabilities of the hydrogel. Thus, conducting controlled modifications that can orientate the structures of both CS and Col to achieve improved polymeric functionality is essential. These polymers are mainly molded to non-allergenic patients. Since the current in vitro and in vivo experiments suggest interesting and motivating results, they usually face significant challenges during clinical tests.

There is still a long path to explore regarding development of new technologies for BTE scaffolds using natural hydrogels, as mentioned previously. Moreover, the industrial scale-up demands innovations to transfer the laboratory results to the supply chain in order to satisfy synthetic processes, such as 3D bioprinting, for the treatment of critical size defects [116,117,118]. Similarly, He et al. [119] developed a CS/acrylamide-based hydrogel using DLP 3D bioprinting, an innovative technology for constructing functional scaffolds for BTE and different tissue engineering applications. Injection molding is a currently advancing trend for the precise construction of biopolymers; nonetheless, the inherent difficulties of handling elevated temperatures and volatile solvents can detriment the biopolymer’s integrity [120,121,122,123,124]. These trending processes intend to biomanufacture scaffolds directly in the living body to improve the tissue viability response and mimic the treated bone defect. The advancement of the research is guaranteed.

## 10. Conclusions

It is interesting to remark on the biomedical and technological advances in designing and evaluating hydrogels for bone tissue engineering. Moreover, herein, we discussed the advances of hydrogel precursors using CS, Col, and gelatin polymers; however, there are only a small number of developments that have been tested on humans. On the other hand, we can highlight critical perspectives that toned to be controlled for the successful application of hydrogels and drug delivery. First, the cross-linking behavior of the polymeric networks plays an important role in achieving the physical parameters, the co-polymer mixtures, and the versatility needed for drug delivery. Second, the chemical characteristics of the polymeric network dictate a substantial reaction under different pH levels, temperatures, and enzymatic conditions that determine the degradation and drug-release rate. Third, the interplay between the polymeric precursor and the filling substances (cross-linkers or inorganic components) is conducted by easily handled hydrogel platforms for minimally invasive and injectable administration methods. Fourth, the application of inorganic components, such as metals and ceramics with counter-ionic polymers, can lead to physical tuning, inherently affecting the matrixes cross-linking and retroactively modulating the drug release level. Therefore, the importance of CS, Col, and gelatin predominantly influences the high biocompatibility and the robust capability to function with different polymers and drugs, tailored osteoblast adhesion, and colonization in the matrix. However, understanding the physical and chemical interaction process occurring among the polymeric materials and delivered drugs is substantial for a better and more reasonable design of BTE hydrogels. Far more important are the fundamental mechanisms governing the cell–material contact interactions, promoting osteoblasts’ growing functionality and the resulting tissue regeneration for optimal bone integration.

## Figures and Tables

**Figure 1 polymers-15-02762-f001:**
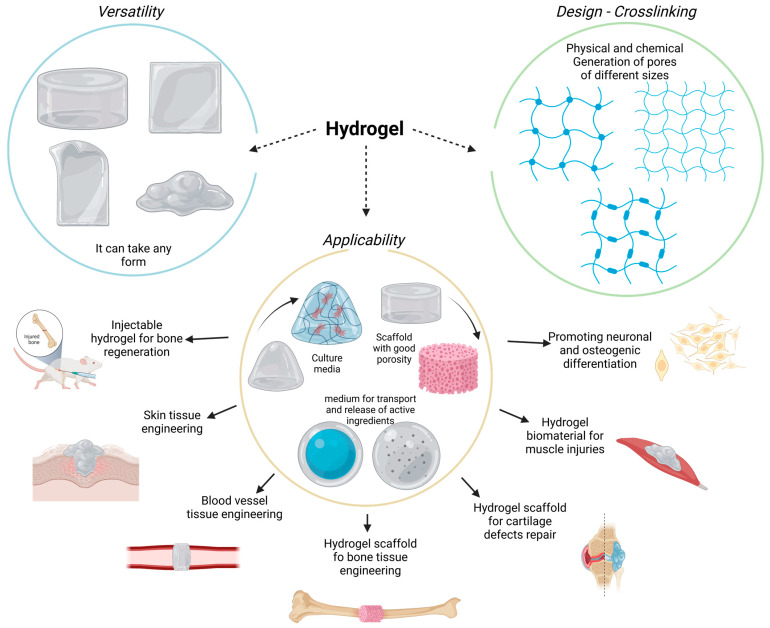
Hydrogels are versatile and malleable biomaterials with several medical applications.

**Figure 2 polymers-15-02762-f002:**
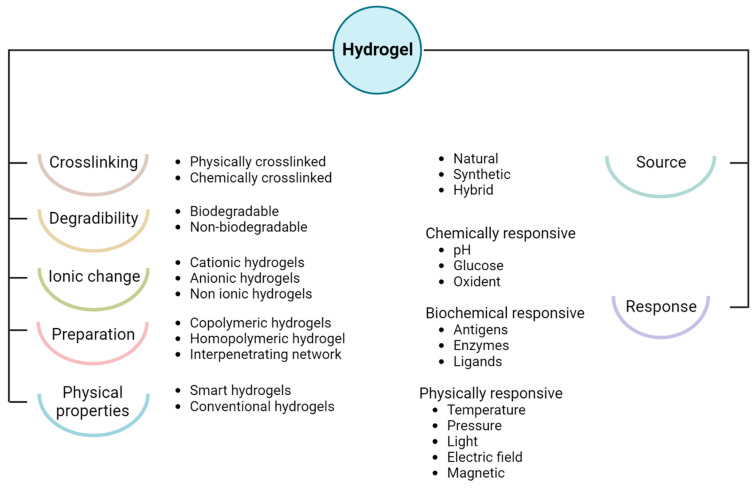
Hydrogel classification according to the physicochemical properties, and applications.

**Figure 3 polymers-15-02762-f003:**
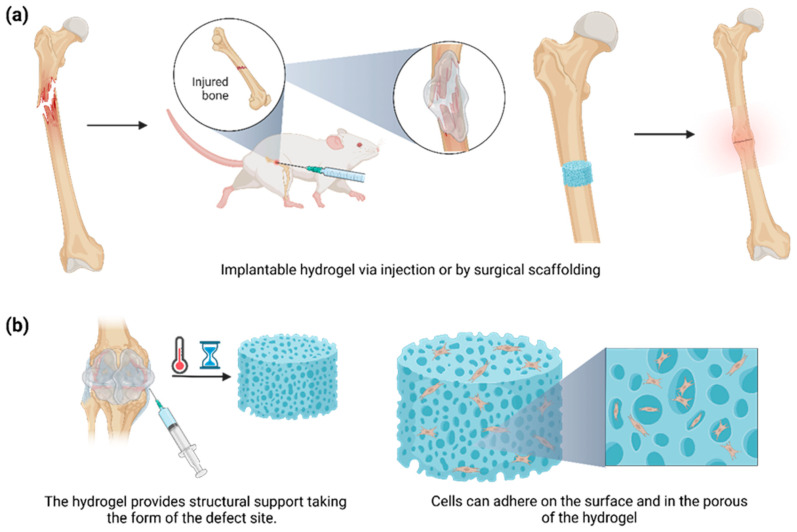
(**a**) Hydrogels can be implanted by injection or surgical scaffolding in the injured area. (**b**) The implantable hydrogel stimuli responsiveness (e.g., temperature and time) can follow the defect’s shape, acting as a fine-tuning platform for promoting bone adhesion and proliferation among the surface matrix and the new tissue.

**Figure 4 polymers-15-02762-f004:**
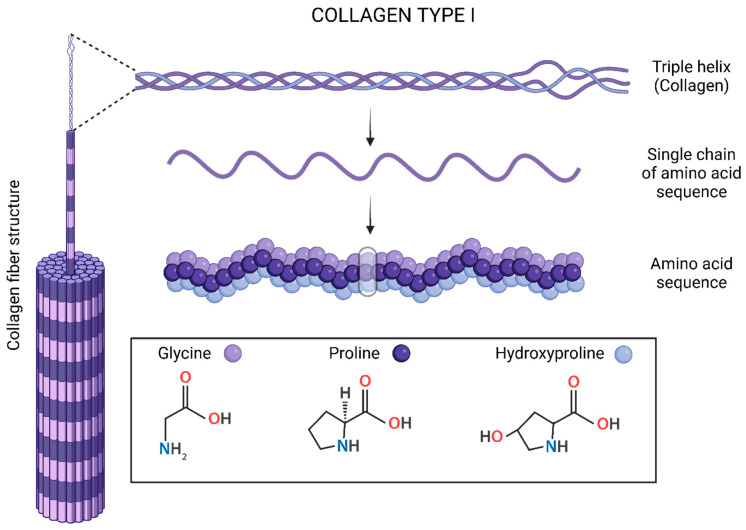
Chemical structure of collagen type I.

**Figure 5 polymers-15-02762-f005:**
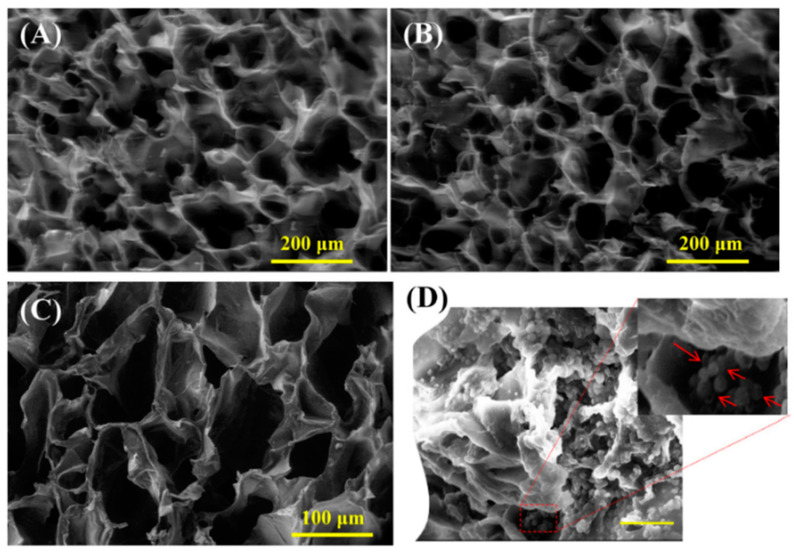
SEM macrographs of hydrogels consisting of (**A**) CS/GP and (**B**) CS/GP/GO. (**C**) Magnified view of CS/GP/GO hydrogel. (**D**) Cells were attached to the pore walls after 4 days of culture. Red arrows indicate the cells. Reproduced and adapted from Saravanan S, Vimalraj S, Anuradha D. Chitosan-based thermoresponsive hydrogel containing graphene oxide for bone tissue repair. Biomed Pharmacother. 2018; 107: 908–917. Copyright © Elsevier Masson SAS. All rights reserved [90].

**Figure 6 polymers-15-02762-f006:**
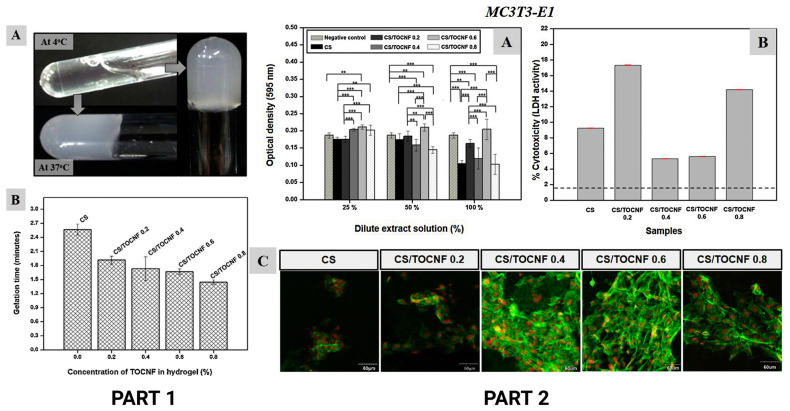
PART 1: (**A**) Optical image of thermosensitive CS/TOCNF hydrogel sol–gel transition with corresponding (**B**) gelation time graph of CS and varying CS/TOCNF hydrogels at 37 °C. PART 2: In vitro biocompatibility determination by MTT assay. LDH cytotoxicity assay and cell proliferation behavior observed through confocal microscopy of pre-osteoblast MC3T3-E1 (**A**–**C**), (*** *p* < 0.001, ** *p* < 0.01). Adapted from Carbodydrate Polymers, Vol. 180, Trang Ho Minh Nguyen, Celine Abueva, Hai Van Ho, Sun-Young Lee, Byong-Taek Lee. In vitro and in vivo acute response towards injectable thermosensitive chitosan/TEMPO-oxidized cellulose nanofiber hydrogel, 246–255, copyright (2017), with the permission of Elsevier [94].

**Figure 7 polymers-15-02762-f007:**
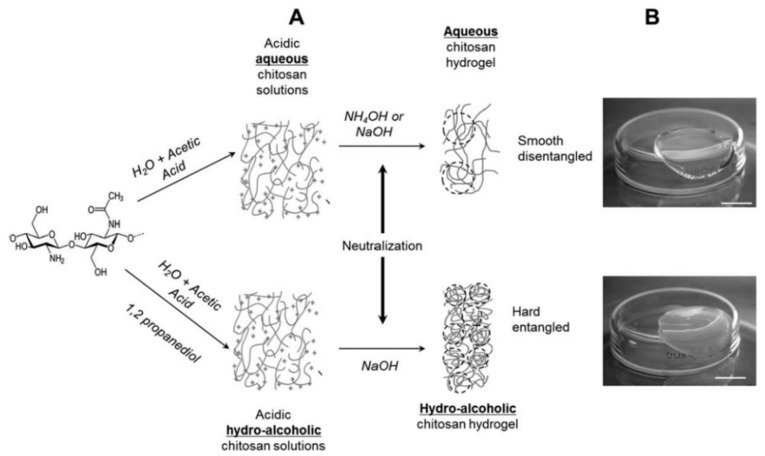
Physicochemical process to synthesize aqueous hydrogels (upper panel) and hydro-alcoholic hydrogels (bottom panel). (**A**) Chitosan dissolution medium; (**B**) photographs of the obtained hydrogels. Dotted circles represent more or less entangled chains. Reprinted with permission of L. Rami, et al., J Biomed Mater Res A. 2014, 102, 10. Copyright © 2013 Society of Plastics Engineers. John Wiley and Sons [95].

**Figure 8 polymers-15-02762-f008:**
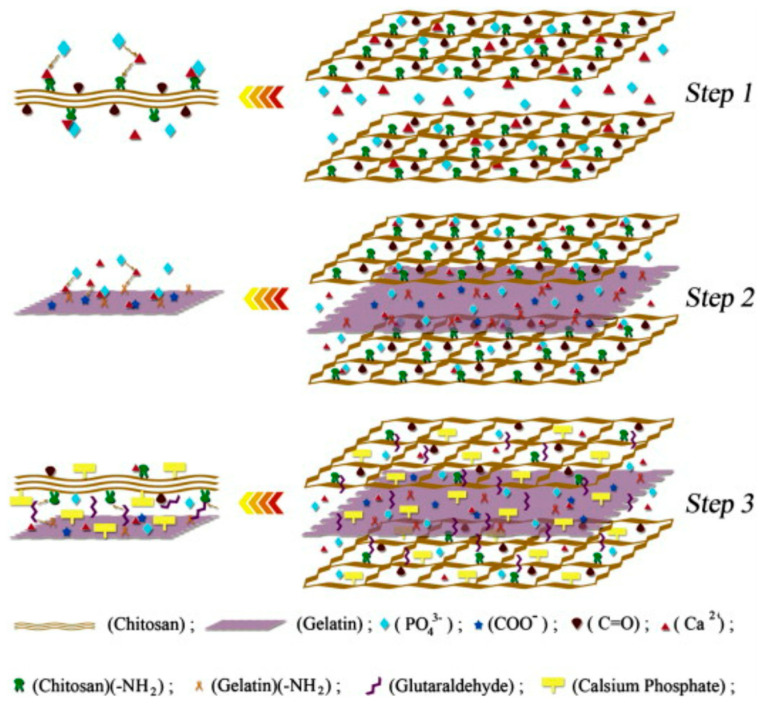
**Top**: The schematic representation of the formation mechanism of homogeneous inorganic/organic composites by in situ precipitation. **Down**: SEM images of Gel-CS/CaP of different organic/inorganic ratios: (**a**) CGC46; (**b**) CGC55; (**c**) CGC64; and (**d**) calibrated EDX area analysis of the composite. Adapted from Materials Science and Engineering: C, Vol. 22, Issue 1, Zahra Babaei, Mohsen Jahanshahi, Sayed Mahmood Rabiee. The fabrication of nanocomposites via calcium phosphate formation on the gelation–chitosan network and the influence of gelatin on the properties of biphasic composites, 370–375, Copyright (2012), with the permission of Elsevier [98].

**Figure 9 polymers-15-02762-f009:**
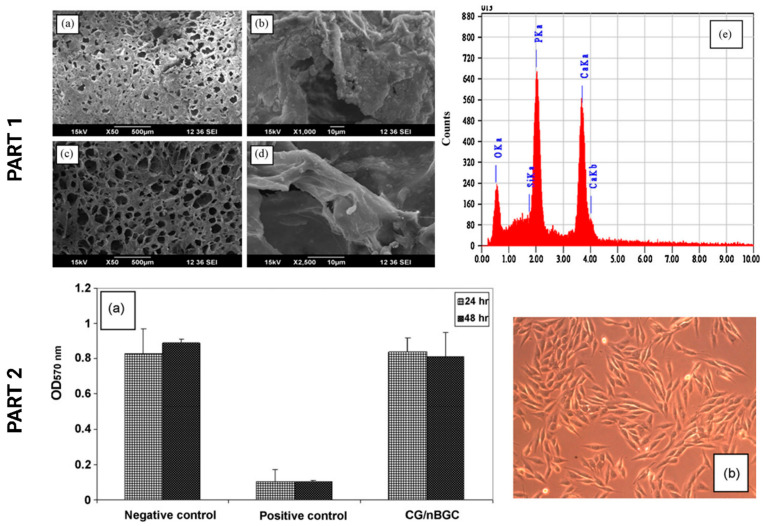
PART 1: (**a**,**b**) In vitro biomineralization studies on the composite scaffolds after 7 days and (**c**,**d**) after 14 days. (**e**) EDS spectra of apatite showed that the Ca/P ratio was 1.64. PART 2: (**a**) MTT assay showing the biocompatibility of the composite scaffolds. (**b**) The morphology of cells grown in direct contact with other cells. Adapted from Chemical Engineering Journal, Vol. 158, Issue 2, Mathew Peter, N.S. Binulal, S.V. Nair, N. Selvamurugan, H. Tamura, R. Jayakumar. Novel biodegradable chitosan–gelatin/nanobioactive glass–ceramic composite scaffolds for alveolar bone tissue engineering, 253–361, Copyright © 2010, with the permission of Elsevier [100].

**Table 1 polymers-15-02762-t001:** Use of different chitosan, collagen, and gelatin formulations for BTE.

Year	Formulation	Model	Effects	Others	Reference
≤2018	Injectable thermosensitive hydrogel based on CS and glycerophosphate.	In vitro	The pores in the hydrogel permit cell infiltration, new tissue ingrowth, nutrient transport, and active ingredient diffusion, improving scaffold functionality. Biocompatible with MSC. GO improves calcium deposition and osteogenesis.	Injectable thermosensitive property. GO addition improves the physico-chemical properties (protein adsorption, swelling capacity, and control of degradable behavior)	[90]
	Scaffold of chitosan–gelatin/nanohydroxyapatite (CSG/nHaP)	In vitro	Improves adhesion and growth of MC3T3-E1 cells compared to nHaP alone. The CSG/nHaP scaffolds exhibited significantly higher cell populations compared to the CS scaffolds with the higher porosity of CSG/nHAP. Superior cytocompatibility of the HaP-containing scaffolds.	Less degradability for the added nHAP Increased mineralization in CSG/nHAP	[92]
	Glutaraldehyde- or genipin-like cross-likers to CS/gelatin scaffolds.	Mouse model femur implantation	CS/gelation with glutaraldehyde has better properties than that with genipin. The scaffold supports the adhesion, viability, proliferation, and osteogenic differentiation capacity of pre-osteoblasts. The scaffold conducts the formation of the extracellular matrix and the expansion of fibroblasts, which produce collagen.	Minimal inflammatory reaction. A homogeneous cell population of CD90, CD105, and CD73, with negative results for CD45 and CD34.	[93]
	Thermosensitive injectable hydrogel of TEMPO-oxidized cellulose nanofibers (TOCNF)/CS	Rat model (Sprague–Dawley male rats)	OCNF enhanced the biocompatibility of hydrogels, both in vitro and in vivo. MC3T3-E1 cells and L929 cells attached to and proliferated on the CS/TOCNF hydrogel better than on the CS hydrogel.	OCNF improved gelation properties. High TOCNF content resulted in faster sol/gel transition, increased porous surface area, and faster degradation. TOCNF hydrogels caused an initial inflammatory response after injection into rats, with the presence of alternatively activated macrophages after 2 weeks.	[94]
	CS with high DA grade and low DA grade of	Rat model	Higher DA CS hydrogels were not suitable for the in vitro culture of hMSC or progenitor-derived endothelial cells. Lower-DA CS hydrogel provided better cell adhesion, tissue regeneration, and neovascularization.	Physical hydrogels prepared from highly DA chitosan were softer and degraded quickly in vivo. Lower-DA CS hydrogel provided a more elastic material, induced a shorter inflammatory response than hydrogel with high DA (20%), and was neutralized by ammonia vapors.	[95]
	Composite scaffolds of chitosan (CS)–gelatin (CG) with bioactive glass–ceramic nanoparticles (nBGC)	In vitro	A higher number of mineral deposits was present on the nano-composite scaffold, which was increased by elevating the incubation time. The nanocomposite scaffolds provided a healthier environment for cell attachment and spreading.	The degradation and swelling behavior of the nanocomposite scaffolds were decreased, while protein adsorption was increased with the addition of nBGC.	[100]
	Gelatin hydrogels incorporating combined stromal cell-derived factor-1 (SDF-1) and bone morphogenetic protein-2 (BMP-2)	Rat model of a critical-sized ulna defect	Enhanced bone regeneration in the presences of SDF-1 and BMP-2 and increased the expression level of chemokine cell-surface receptor-4 (Cxcr4), runt-related factor-2 (Runx2), and osteocalcin genes. Promoted a vascular-like structure. The combined release of SDF-1 and BMP-2 enhanced the recruitment of osteogenic cells and angiogenesis.	-	[103]
	Photopolymerizable hidrogel with methacrylated glycol chitosan and semi-interpenetrating collagen with a riboflavin photoinitiator under blue light.	In vitro	Enhanced cellular attachment, spreading, proliferation and osteogenic differentiation of BMSCs seeded on the hydrogels compared to those without Col hydrogels. The mineralization was increased.	Col enhanced the compressive modulus and slowed the degradation rate of the hydrogels.	[107]
	CS/HA and CS/alginate hydrogels laden with MC3T3-E1 and processed by 3D bioprinting	Rat model of calvaria bone defects	The hydrogels maintained cell viability and proliferation after printing. The CS/HA hydrogel had peak expression levels for early- and late-stage osteogenic markers. CS and CS/HA hydrogels were mineralized and differentiated after 21 days of culture.	It was shown for the first time that CS and HA can be mixed with cells and printed successfully. The tested groups had viscoelastic properties. CS and CS/HA were stable under physiological stimulation.	[111]
2019	CS-sulfonated graphene oxide and CS-graphene oxide scaffold	Mouse model femur implantation	Bone regeneration and increased biocompatibility using graphene oxide. Healthy cellular viability and dense bone morphology.	Well-controlled drug delivery. Antibacterial effect against *S. aureus.*	[5]
	Collagen/tacrolimus hydrogel	Rat model of calvaria bone defects	The collagen hydrogel contained 1000 μg of tacrolimus, which was adequate in terms of cell proliferation. In vivo studies provided evidence of the potential of the developed hydrogel for bone healing.	Highly porous structure with interconnected pores Hydrogel showed appropriate swelling, drug release, and blood compatibility behavior.	[34]
	Collagen/CS electrospun nanofiber membranes (ECCMs)	Rat model of calvaria bone defects	Better cell proliferation and biocompatibility. Expression of BALP and OC showed that a higher level of osteogenic activity existed in the ECCMs group than other groups at both of the early and late stage. Newly formed bone almost fully filled the cranial defects in the ECCM group after 8 weeks.	Stronger tensile strength was achieved by the ECCMs, in addition to a lower and more stable degradation rate of the membrane. A highly porous and nanofibrous electrospinning structure was present.	[97]
	TiO_2_/gelatin–chitosan hydrogel	In vitro	TiO_2_-loaded gelatin/chitosan hydrogel showed higher adhesion than gelatin/chitosan expression of osteocalcin and F-actin proteins. Higher mineralization and alkaline phosphatase response.	The addition of TiO_2_ nanoparticles showed good thermal stability on the hydrogel.	[99]
	Composite scaffold of CS/Gel with glutaraldehyde-like cross-linker	Rat model of calvaria bone defects	Supported cell viability and proliferation. Extensive formation of a nHA phase. The concentration of glutaraldehyde significantly affected the expression of specific osteo/odontogenic genes. The CS/Gel scaffold type demonstrated a better biological response.	Increased degradability with 0.1% of glutaraldehyde.	[113]
2020	Catecol–chitosan (CA-CS) hydrogels functionalized with zeolitic imidazolate frameworks: 8 nanoparticles (ZIF-8) (CA-CS/Z)	Rat model of calvaria bone defects	Enhanced adhesion and led excellent biocompatibility, osteogenesis, and promotion of bone regeneration. Enhance paracrine of the vascular endothelial growth factor (VEGF). The ZIF-8 NPs released from the hydrogels were also able to up-regulate the production and secretion of ALP, COL 1, and osteocalcin markers, promoting the osteogenic differentiation of rBMSCs.	The bone transplant environment was stabilized. Hydrogels exhibited advanced rheological properties and reliable mechanical strength. Antibacterial activity was present against *S. aureus* and *E. coli*.	[102]
	Thermosensitive hydrogel/nanoparticle system made of CS and glycerol phosphate loaded with vancomycin NPs (VCM) (VCM/Gel)	Rabbit model of chronic osteomyelitis	The VCM-NPs/Gel promoted osteoblast proliferation. The VCM-NPs/Gel showed excellent anti-infection properties and accelerated bone repair under osteomyelitis conditions.	The VCM-NPs had high encapsulation efficiency and drug loading. The VCM-NPs/Gel exhibited sustained release of VCM over 26 days. Antibacterial activity was present against *S. aureus*.	[114]
2021	CS/HA	In vitro	CS/HA scaffolds supported cell proliferation and differentiation. Scaffolds with higher concentrations of HA (60 and 70%) showed an impressive effect on osteogenic differentiation of hMSC towards a mature osteoblast phenotype.	Prevented degradation in an in vitro cell culture model as well as pro-inflammatory events. Showed a good effect on the expression of anti-inflammatory cytokines (IL-10 and IL-4); meanwhile, it was able to decrease pro-inflammatory cytokine (TGF-β) levels.	[96]
	AgNPs/Gelatin	In vitro	AgNPs/Gel hydrogels are nonhazardous to osteoblasts. Improved survival and spreading of osteoblasts cells on the hydrogel were achieved.	-	[106]
	Injectable CS/Col hydrogel using carboxylic acid functionalized single-walled carbon nanotubes (COOH-SWCNTs) as integrators and sodium β-glycerophosphate (β-GP) salt as a cross-linker.	In vitro	The evaluated hydrogels formed a layer of HA on the surface. Hydrogels are non-toxic, increasing cell proliferation and osteogenic differentiation compared to pure CS/Col.	Thermoresponsive with sol–gel transition occurring at physiological temperature. Degradation and swelling properties were found to be composition-dependent. Optimal injectable properties. Enhanced significant mechanical properties.	[108]
2022	Collagen hydrogel nanocomposite in combination with 2% strontium (Co/BGSr2%)	Full-thickness bone defect regeneration in the rabbit animal model	The tests (radiographical and histopathological) showed better results, highlighting the Co/BGSr2% + MSCs group. The highest expression level of osteocalcin was detected in Co/BGSr2% + MSCs, especially at the fourth week of post-transplantation.	Large pores.	[109]
2023	Scaffolds based on Cs/Alg/HD using the direct ink writing 3D printing technique.	In vitro	The addition of HD particles had a positive effect on cell viability and attachment.	Cs/Alg/HD70 demonstrated the highest yield strength (1.38 MPa) and elastic modulus (125.71 MPa), as well as the highest degradation rate and the lowest contact angle.	[112]
	nHA was incorporated into a dopamine-modified gelatin (Gel-DA) hydrogel system using a polydopamine-like cross-linker	Rat femoral defect model	The hydrogel had excellent cell adhesion and proliferation, leading to the improved biocompatibility. The hydrogel accelerated the bone repair efficiency in in vivo tests.	Adding polydopamine-functionalized nHA increased the compressive strength of the Gel-Da hydrogel. The gelation time of the Gel-DA hydrogels with PHA was controllable.	[115]

## Data Availability

Not applicable.

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
