# Peer review of "Chitosan, Gelatin, and Collagen Hydrogels for Bone Regeneration"

_polymers, 2023, doi:10.3390/polym15132762_

Round 1
Reviewer 1 Report
Hydrogels are versatile biomaterials characterized by three-dimensional cross-linked highly hydrated polymeric networks. These polymers exhibit great variety of biochemical, biophysical properties and promising candidate for medical applications. However, there are still some issues that need to addressed:
1. In the introduction part line 66 and 67, you are discussing about novel technologies and procedures but later on you have not mentioned any procedure.
2. Figure 7 caption is need to be revised.
3. Please eliminate the use of redundant words. E.g. In this way, Recently, Respectively, therefore, currently, thus, hence, finally, to do this, first, in order, however, moreover, nowadays, today, consequently, in addition, additionally, on the other hand, furthermore. Please revise all similar cases.
4. Page 1, line 37-39, Sentence ʺFar more critical is the difficulty of repairing significant bone defects…ʺ need to be revised.
5. The structure of this manuscript should be re-organized.
6. The main findings and novelty of this manuscript should be highlighted both in the abstract and conclusion part.
7. Please use some transition words and topic sentences between different paragraphs to make it easier for the reader to follow.
8. English writing needs substantial improvement.
9. Abstract should be unique and summarize the whole manuscript. But you have copied the lines 14-19 (abstract) from Page 2, lines 77-83.
10. Figure 1 is too general. Please be specific if you are mentioning properties or applications.
11. Sentences Page 4 Lines 140-144 (hydrogels for bone regeneration) are same as on Page 2 Line 60-65(introduction). Please revise these sentences.
12. Sentence Page 6 line 223 ʺInterestingly, cross-linked CS hydrated polymers are classified on theʺ need to be revised.
13. Please write the correct spell of ʺsol-gelʺ on page 6 line 229.
14. Authors should explain the collagen Type II and Type III on page 7 line 239.
15. Sentence ʺInterestingly, cross-linked CS hydrated polymers are classified on the…ʺ on page 9 line 316 need to revise.
16. Explain the full form of abbreviation ʺMC3T3-E1ʺ, ʺTEMPOʺ, ʺrBMSCsʺ, ʺMSCsʺ.
17. Figure 6 graphics are very poor. Please improve the figure quality.
18. Sentence ʺIt has been reported that inorganic content in polymeric composite materials can…ʺ on page 12 line 395 has no clear information.
19. Authors should the correct the symbols like ʺCa2+ and PO43-ʺ to Ca 2+ and PO 4 3- , ʺTiO2ʺ to TiO 2 ,
20. Graphs present in Figure 8 are of low quality. Please enhance the quality of figures.
21. In this paper you have not mentioned any specific procedure to produce hydrogels from biopolymers.
22. Authors should add tables which reflects the comparison results biopolymers-based hydrogels and synthetic hydrogels.
23. Reference style should be same. Reference 1 has different style.
24. Reference 42, 58, 59, 62, 76, 78 and 81 have missing
English needs revision, already few typo mistakes have been mentioned.
Author Response
Dear reviewer, thank you very much for your important recommendations and the experience shared in your comments; we much appreciate it. As per your comments, we have addressed the point itemized below. We believe that the present modifications would fulfill your essential indications. We hope you find our article as interesting as we do.
Hydrogels are versatile biomaterials characterized by three-dimensional cross-linked highly hydrated polymeric networks. These polymers exhibit great variety of biochemical, biophysical properties and promising candidate for medical applications. However, there are still some issues that need to addressed:
- In the introduction part line 66 and 67, you are discussing about novel technologies and procedures but later on you have not mentioned any procedure.
Thank you for your recommendation. The procedures have been mentioned as per your recommendation.
- Figure 7 caption is need to be revised.
The figure description has been modified.
- Please eliminate the use of redundant words. E.g. In this way, Recently, Respectively, therefore, currently, thus, hence, finally, to do this, first, in order, however, moreover, nowadays, today, consequently, in addition, additionally, on the other hand, furthermore. Please revise all similar cases.
The manuscript was modified by reducing redundant words and conjunctive adverbs.
- Page 1, line 37-39, Sentence ʺFar more critical is the difficulty of repairing significant bone defects…ʺ need to be revised.
The statement has been revised.
- The structure of this manuscript should be re-organized.
Thank you for this recommendation. We restructured the manuscript we hope you find it logically organized and assimilable.
- The main findings and novelty of this manuscript should be highlighted both in the abstract and conclusion part.
The main findings and novelty of our manuscript were highlighted in the abstract and conclusion sections.
- Please use some transition words and topic sentences between different paragraphs to make it easier for the reader to follow.
We improved the manuscript including transition words.
- English writing needs substantial improvement.
Thank you, the English draft has been revised.
- Abstract should be unique and summarize the whole manuscript. But you have copied the lines 14-19 (abstract) from Page 2, lines 77-83.
The abstract has been modified and improved; we appreciate the recommendation.
- Figure 1 is too general. Please be specific if you are mentioning properties or applications.
Figure 1 was improved, supporting illustrations regarding wide applications.
- Sentences Page 4 Lines 140-144 (hydrogels for bone regeneration) are same as on Page 2 Line 60-65(introduction). Please revise these sentences.
The sentences have been revised.
- Sentence Page 6 line 223 ʺInterestingly, cross-linked CS hydrated polymers are classified on theʺ need to be revised.
The structure of the sentence was modified to improve coherence.
- Please write the correct spell of ʺsol-gelʺ on page 6 line 229.
We have corrected it.
- Authors should explain the collagen Type II and Type III on page 7 line 239.
We included a profound explanation and the importance of Type II and Type III collagen.
- Sentence ʺInterestingly, cross-linked CS hydrated polymers are classified on the…ʺ on page 9 line 316 need to revise.
It was revised.
- Explain the full form of abbreviation ʺMC3T3-E1ʺ, ʺTEMPOʺ, ʺrBMSCsʺ, ʺMSCsʺ.
The descriptions have been included; thank you for this observation.
- Figure 6 graphics are very poor. Please improve the figure quality.
We have improved the quality of Figure 6.
- Sentence ʺIt has been reported that inorganic content in polymeric composite materials can…ʺ on page 12 line 395 has no clear information.
The sentence was modified to clarify the context of the topic.
- Authors should the correct the symbols like ʺCa2+ and PO43-ʺ to Ca 2+ and PO 4 3- , ʺTiO2ʺ to TiO 2 ,
Thank you for this recommendation. However, we consider that the original symbols follow the statements of IUPAC. Any way your recommendations were taken in account as you can see in the corrected text. Therefore, we very much appreciate to conserve the original symbols.
- Graphs present in Figure 8 are of low quality. Please enhance the quality of figures.
The quality has been enhanced.
- In this paper you have not mentioned any specific procedure to produce hydrogels from biopolymers.
This is an important observation. New statements regarding fabrication and applications have been included to improve the manuscript.
- Authors should add tables which reflects the comparison results biopolymers-based hydrogels and synthetic hydrogels.
Thank you for your recommendation. However, the objective of the work is to review state of the art regarding the science and technology of the natural polymer chitosan, collagen, and gelatin for BTE. Therefore, we consider that comparing synthetic vs. natural polymers will dissuade the attention of the original aim of the work. Besides, we have added a new table with the most relevant works about the reviewed biopolymers.
- Reference style should be same. Reference 1 has different style.
We modified it.
- Reference 42, 58, 59, 62, 76, 78 and 81 have missing
Thank you, we reviewed the references.

Reviewer 2 Report
This is quite comprehensive review of the Chitosan, gelatin and collagen hydrogels for bio-applications.
This review covers a recent developments and summarized all important data. The figures provided preserved copyrights and they supportive. The whole report is well-written and welm organized.
I recommend to accept it.
English is good
Author Response
Comments and Suggestions for Authors
This is quite comprehensive review of the Chitosan, gelatin and collagen hydrogels for bio-applications.
This review covers a recent developments and summarized all important data. The figures provided preserved copyrights and they supportive. The whole report is well-written and welm organized.
I recommend to accept it
Comments on the Quality of English Language
English is good
Dear reviewer, thank you very much for your time and effort in revising our manuscript.

Reviewer 3 Report
1. resolution of microscopical examination needs to be improved
2. the limitations, adverse effects and problems of hydrogel scaling up need to be addressed.
The review needs extensive English revision
Author Response
Dear reviewer, we appreciate your important recommendations and the experience shared in your comments. As per your comments, we have addressed the point itemized below.
Comments and Suggestions for Authors
- resolution of microscopical examination needs to be improved
We improved the quality of the Figures 8 and 9.
- the limitations, adverse effects and problems of hydrogel scaling up need to be addressed.
Information regarding the limitations and future perspectives has been included.
Comments on the Quality of English Language
The review needs extensive English revision
Thank you. The English style was revised.

Reviewer 4 Report
Some concerns should be addressed as follows:
1. The authors should add a table, illustrating the different hydrogels fabricated in previous studies based on chitosan, collagen and gelatin for bone treatment. Moreover, significant findings should be highlighted in this table.
2. Hydrogels with antibacterial and immunomodulatory capabilities should be discussed.
3. Future perspectives should be presented.
4. Fig. 8 should be improved.
5. The authors should incorporate articles recently published in 2023. For instance, https://doi.org/10.3390/pharmaceutics15030807; https://doi.org/10.1016/j.ijpharm.2023.122649; https://doi.org/10.3389/fphar.2023.1050954; https://doi.org/10.3390/ijms24043744; https://doi.org/10.1021/acsbiomaterials.2c00730; https://doi.org/10.1016/j.ijbiomac.2023.124339).
Author Response
Dear reviewer, thank you very much for your essential observations and the experience shared in your comments; we much appreciate it. As per your recommendations, we have addressed the points itemized below. We believe that the present modifications would fulfill your essential indications. We hope you find our article as interesting as we do.
Comments and Suggestions for Authors
Some concerns should be addressed as follows:
- The authors should add a table, illustrating the different hydrogels fabricated in previous studies based on chitosan, collagen and gelatin for bone treatment. Moreover, significant findings should be highlighted in this table.
Following your recommendations, a new Table (Table 1) has been incorporated into the manuscript.
- Hydrogels with antibacterial and immunomodulatory capabilities should be discussed.
These important topics are appropriately discussed in the manuscript.
- Future perspectives should be presented.
We include information discussing future perspectives and current limitations.
- Fig. 8 should be improved.
We have improved the quality of Figure 8.
- The authors should incorporate articles recently published in 2023. For instance, https://doi.org/10.3390/pharmaceutics15030807; https://doi.org/10.1016/j.ijpharm.2023.122649; https://doi.org/10.3389/fphar.2023.1050954; https://doi.org/10.3390/ijms24043744; https://doi.org/10.1021/acsbiomaterials.2c00730; https://doi.org/10.1016/j.ijbiomac.2023.124339).
Thank you for the recommendation; we have incorporated the references.

Round 2
Reviewer 4 Report
The authors addressed all the previous claims carefully.